# Structure and Properties of the Xerogels Based on Potassium Silicate Liquid Glass and Urea

**DOI:** 10.3390/molecules28145466

**Published:** 2023-07-17

**Authors:** Alexander Gorokhovsky, Igor Burmistrov, Denis Kuznetsov, Alexander Gusev, Bekzod Khaidarov, Nikolay Kiselev, Elena Boychenko, Evgeny Kolesnikov, Ksenia Prokopovich, Yuri Konyukhov, Maksim Kravchenko

**Affiliations:** 1Department of Functional Nanosystems and High Temperature Materials, National University of Science and Technology (MISIS), 119049 Moscow, Russia; dk@misis.ru (D.K.); nanosecurity@mail.ru (A.G.); bekzod1991@mail.ru (B.K.); nikokisely12345@gmail.com (N.K.); kea.misis@gmail.com (E.K.); prokopovitchksenia@yandex.ru (K.P.); ykonukhov@misis.ru (Y.K.); 2Moscow Power Engineering Institute, National Research University, 111250 Moscow, Russia; kravchenkomv@mpei.ru

**Keywords:** liquid glass, urea, crystallization behavior, xerogels, structure, water resistance

## Abstract

The xerogels based on the aqueous solutions of urea in potassium silicate liquid glass (PSLG) were produced by CO_2_ bubbling and investigated. The structure and chemical composition of the obtained materials were analyzed. Using the SEM, XRD, IR-FT, DSC, and low energy local EDS analysis, it was recognized that the dried gels (xerogels) contained three forms of urea: oval crystals of regular shape appeared onto the surface of xerogel particles; fibrous crystals were located in the silicate matrix; and molecules/ions were incorporated into the silicate matrix. It was shown that an increase in [(NH_2_)_2_CO] in the gel-forming system promoted increased contents in crystalline forms of urea as well as the diameter of the fiber-shaped urea crystals. A rate of the urea release in water from the granulated xerogels containing 5.8, 12.6, and 17.9 wt.% of urea was determined by the photometric method. It was determined that the obtained urea-containing xerogels were characterized with a slow release of urea, which continued up to 120 days, and could be used as controlled release fertilizers containing useful nutrients (N, K).

## 1. Introduction

The use of nitrogen-containing fertilizers is very important in modern agricultural production. However, more than half of nitrogen containing fertilizer is released into the environment with negative impacts on air, water, and soil quality. Nitrogen loss to the environment contributes to greenhouse gas emissions and climate changes [1].

The best way to solve this problem involves an introduction of the traditional nitrogenous fertilizer (KNO_3_, urea) into an oxide matrix. This approach would significantly increase the efficiency of fertilizers, reduce the amount of them applied to the soil and, as a result, decrease air and groundwater pollution [1,2,3,4,5,6].

Some complex vitreous fertilizers based on alkali silica and phosphate glasses forming a matrix to incorporate different nutrients are known [3,6,7]; however, it was impossible to introduce nitrogen in their structure due to high levels of their fusion temperature (T > 1000 °C) promoting thermal decomposition of nitrogen-containing compounds (ammonium salts, nitrates, or urea). An introduction of nitrogenous compounds in the porous matrix of silica–phosphate glasses [8,9] does not allow for resolving the above-mentioned problem and has a high cost.

It is known that xerogels based on alkali-silicate liquid glasses are sensitive to outside conditions. Additionally, the xerogels based on alkali–silicate liquid glasses and containing less than 15% of acid solidifier have relatively low water resistance. To increase the water resistance, the liquid glass compounds are modified with other solidifiers such as urea [10,11]. Furthermore, it was shown that the alkali–silicate glasses containing urea as solidifier could be characterized with an acceptable water resistance, which would allow their use as nitrogenous containing slow-release fertilizer. However, an acceptable water resistance of the dried gels based on this system occurred at relatively low contents of urea (less of 5 wt.%). It was also recognized [12] that CO_2_ bubbling could reduce the pH of such solutions and support gelation in the potassium–silicate colloidal solutions. The precipitates obtained by CO_2_ bubbling through the urea–alkali silicate colloidal solution (24 h) were separated from the residual aqueous solutions and dried to produce the xerogels. The data of IR-FT spectroscopy indicated the presence of urea in these xerogels but the content of urea in the obtained xerogels was not determined. It was noted only that the final product contained urea incorporated in the silicate matrix in the molecular form, taking into account the absence of any crystalline urea reflexes in the XRD patterns.

It was also determined [13] that the rate of gelation in the system of sodium silicate aqueous solution–urea depends on a temperature and ratio of the components. The bulk gelation experiments indicated that a gelation time decreased with increased urea concentrations. The relationship between the gelation time and sodium silicate concentration exhibited a minimum at approximately 7 wt.% sodium silicate when the urea concentration was 3.6 wt.%. It was also established that the presence of calcite (CaCO_3_) supported increased stability in the obtained gels.

Taking into account the above-mentioned results, the aim of this research was related to the specification of the mechanism of the processes in the system of urea–alkali–silicate liquid glass as well as the determination of the chemical composition of this system, which allows obtaining dried gels, characterized with high content of urea and moderate water resistance. Such combination could be useful to consider the obtained xerogels as promising slow-release nitrogenous fertilizer.

## 2. Results

### 2.1. Gels Characterization

The materials obtained after the drying of the gelated systems at 60 °C were powders consisting of various sized particles (Figure 1). The large sized particles of the dried gels with high contents of urea were incrusted with urea crystals.

The data on pore distribution, density, porosity, and specific surface area of the obtained xerogels are reported in Table 1 and Figure 2.

Taking into account the obtained results, we could classify the obtained dried gels as xerogels. These materials are produced by drying at atmospheric pressure under rather severe conditions, leading to the collapse of macropores and a significant increase in the density of the products (Table 1). It is important that, in contrast to aerogels, which have a very high specific surface area (higher than 500 m^2^/g) and low density (less than 0.5 g/cm^3^) [14,15], the density of dried urea-containing gels is close to the density of monolithic xerogels based on pure alkali–silicate glasses [16]. Their porosity (17–20%) also is close to the porosity of silicate xerogels (up to 25% [16]) as well as their surface area (139–158 m^2^/g) and small pore size (5–10 nm). 

The chemical composition of the dried gels (xerogels) produced with the admixtures of 10, 20, and 30 g of urea to 100 mL of the potassium silicate liquid glass (PSLG) are noted in Table 1. These compositions were estimated, taking into account the quantities of components used and the data of TGA (reported later in this text). It is possible to note that an increase in urea content in the xerogel increased the water content, especially in the gel No3.

### 2.2. Structural Features of Xerogels

The XRD patterns of the PSLG based xerogels, obtained with different admixtures of urea by drying at 60 °C for 48 h, are reported in Figure 3a. The obtained data shows that urea is represented in the structure of the xerogels in various crystalline forms. The main XRD reflection of urea (100) has a bifurcated shape (maximums at 21.87 and 22.05°). Furthermore, this reflection of the crystalline urea incorporated into the silicate–xerogel structure shifted in the lower 2θ-angles range in comparison with pure crystalline urea [17,18] or crystalline urea mechanically mixed with the urea-free xerogel (Figure 3b, maximum at 22.4°). The XRD patterns of the obtained xerogels indicate that a quantity of planes and d-values of urea, incorporated into the silicate matrix, changed in comparison with pure crystalline urea, similar to [18]. This effect may occur due to the overlapping of the planes of urea crystals and reorientation in the structure due to the different morphology of the crystals, which arose under the influence of the silicate–xerogel matrix.

It can also be noted that the XRD pattern of the mechanical mixture of the urea-free potassium silicate gel produced only by gelation of liquid glass by CO_2_ bubbling has the K_2_CO_3_ reflections, which are absent in the xerogels based on the PSLG-urea solutions (Figure 3b).

To estimate the fraction of urea represented in the obtained xerogels in the crystalline and non-crystalline forms, an intensity of the main (110) urea reflection (at 21–22°) was measured for the systems: (1) mechanical mixtures obtained with the powdered urea-free xerogel produced with PSLG(+CO_2_) (100 mL PSLG) and crystalline urea (10, 20, and 30 g) (System 1), and (2) xerogels based on the PSLG-urea(+CO_2_) system (containing 10, 20, and 30 g of urea in 100 mL of PSLG) (System 2). The obtained data are reported in Figure 4a. 

The intensity of the main reflection of (NH_2_)_2_CO for the mechanical mixtures (System 1) has a linear dependence on urea content (line 1 in Figure 4a). However, for the xerogels based on the PSLG-urea solutions (System 2), this dependence has a non-linear character (line 2 in Figure 4a), and the intensities of the (110) reflection for the xerogels obtained in the System 2 are much lower. It is possible to also note that an increase in the urea content promotes increased fraction of the crystalline urea, which can be estimated to 6.7, 15.2, and 74.1 mol.% for the products containing 6.8, 12.6, and 17.9 wt.% of urea, respectively, taking into account the data of Figure 4a. 

Thus, (NH_2_)_2_CO exists in the structure of the obtained xerogels in the crystalline and non-crystalline forms.

The IR spectra obtained by FT-IR spectroscopy with the xerogels containing different quantity of urea are reported in Figure 4b. The absorption bands of pure urea [19,20] are identified in Figure 4b in the spectral range (1400–1900 cm^−1^), which is free of the absorption bands of different silicates [21]. It is important that the symmetric deformation vibration band δ_s_(NH_2_) for the urea containing xerogels was blue-shifted to 1683 cm^−1^ compared to that in pure crystalline urea (1668 cm^−1^). This shift indicates the interaction of NH_2_ groups of urea with Si-O-H groups of the silicate matrix similar to the case considered in [22] and has been mentioned earlier for the IR spectra of R-NH_3_^+^ groups in the organic substances [23]. 

Some structural features of the xerogel particles can be analyzed taking into account SEM data and low energy EDS local microanalysis, reported in Figure 5.

The xerogels, solidified by drying the gels based on aqueous solutions of urea in the PSLG treated with CO_2_ bubbling, have porous structure with inclusions of needle-shaped crystals of urea with a diameter of 2–5 µm and length of 100–150 μm (Figure 5). A quantity of such crystals increases with the content of urea in the gel forming mixture. In addition, the investigated xerogels are characterized with a presence of large-scale oval-shaped urea crystals localized on the surface of xerogel particles (Figure 1a and Figure 5e). Closed spherical pores in the xerogel structure have a diameter of 2–8 µm (Figure 5b,e). It is necessary to note that the surface of these pores is coated with a thin layer of crystals, more likely K_2_CO_3_, which could be formed as a result of CO_2_ interaction with H_2_O and K^+^ during the bubbling and drying processes. The chemical composition of the xerogel matrices obtained using the low energy EDS analysis shows the presence of nitrogen (N) and carbon © and indicates the existence of urea in the crystalline and in non-crystalline (molecular or ionic) form. 

### 2.3. Thermal Behavior

The DSC data obtained for the xerogels based on potassium–silicate liquid glass and produced with and without urea admixtures, are reported in Figure 6 (curves 2, 3, and 1, respectively) in comparison with the DSC curve of pure crystalline urea (curve 4).

It can be noted that the DSC covers of the xerogels, based on the system of potassium–silicate liquid glassurea, differ from the DSC covers of free crystalline urea and xerogel based on PSLG treated by CO_2_ bubbling without urea additives. The endothermal peaks at 118 and 166 °C (marked as a and e in Figure 6) marked in the DSC curve of urea-free xerogel (curve 1) and corresponding to the desorption of physically adsorbed [24] and structural water [25], respectively, change their location in the DSC covers of the xerogels based on the system of PSLG-urea. The first peak (a) with the addition of urea shifts to the lower temperature range (from 118 to 78 °C) and its intensity decreases, while the second peak (e) shifts in the range of higher temperatures (from 166 to 184 °C). 

On the other hand, the endothermal peak at 143 °C (marked as b in Figure 6), considered as the one corresponding to the melting of crystalline urea [26], bifurcates in the DSC curves of the xerogels based on the system of PSLG-urea (the peaks marked as b and b’ in Figure 6). 

Weight losses of xerogels allow for an estimation of the level of [H_2_O] present in the samples No 2 and 3 (Table 1) and indicate an increase in the physically adsorbed water content (corresponding to the range of 25–180 °C) in the xerogel with higher concentrations of urea. 

It is necessary to note also that the endothermal peaks at 232 and 247 °C (Figure 7) appearing in the DSC covers of pure crystalline urea and traditionally contributed to the result of thermal decomposition of molten urea and its derivatives (biuret, cyanuric acid, ammelide) formed at lower temperatures [26], are much less intensive in the DSC covers of urea containing xerogels and shift in the range of lower temperatures.

### 2.4. Chemical Durability

The kinetic data indicating a rate of urea release from the xerogels obtained with various admixtures of urea are reported in Figure 7.

Taking into account that the urea content was the same in both of the investigated systems (No 2 and 3, Table 1), and similar pH values of the obtained aqueous dispersions, it is possible to note that the xerogels containing 12.6 and 17.9 wt.% of urea were characterized with much slower release of (NH_2_)_2_CO. In the initial stage of soaking in H_2_O, the xerogel containing 17.9 wt.% of urea has a little bit faster release of this substance in water due to the presence of large oval-shaped urea crystals onto the surface of xerogel particles; however, in the following period, when a rate of release is determined by degradation of the silicate matrix, this characteristics are similar to both urea containing xerogels.

## 3. Discussion

The obtained results indicate that an introduction of urea in the potassium silicate liquid glass supports intensive gelation of the investigated system and allows minimizing the time of its treatment by CO_2_ bubbling to complete this process. 

The xerogels obtained by drying the urea-containing gels are almost free of K_2_CO_3_ (Figure 3), however, contain higher quantity of H_2_O (Table 1). In accordance with the data of SEM, it is possible to propose that small-sized K_2_CO_3_ crystals are located, mostly, on the surface of spherical micro-pores formed by bubbling the PSLG-based aqueous solutions with CO_2_ (Figure 5e). 

The obtained xerogels contain some quantity of molecular water (Table 1), at the same time, in comparison with urea-free xerogels, the physically adsorbed H_2_O molecules are less strongly connected with the silicate matrix of urea-containing xerogels (the exo-peak marked as a in Figure 6b is shifted in the range of lower temperatures), whereas the structural water of these xerogels is more strongly fixed in the matrix in comparison with the urea-free xerogel.

It is very important that all the obtained results indicate the presence of urea in various structural forms appearing at different stages of the gelation and drying the gels.

In the initial stage of the gelation process, taking place in the urea-PSLG system, the (NH_4_)_2_CO molecules interact with Si-O-H groups of the silicate chains present in the colloidal solution.
|     H_2_O   |
O-Si-O^−^ (^+^NH_3_CONH_3_^+^)^−^O-Si-O
|     H_2_O   |

This process favors gelation of the system. At the same time, the additional treatment with CO_2_ allows completing the gel formation due to increased pH values [12,13].

The following drying of the gels allows obtaining the xerogels. The structural features of these materials depend on urea contents. In the xerogels containing up to ~6 wt.% of urea, all the (NH_3_)_2_CO molecules containing in the PSLG aqueous solution can be incorporated into the structure of xerogels in the ionic form (^+^NH_3_CONH_3_^+^) and stabilize the silicate–xerogel matrix.
|           |
O-Si-O^−^ (^+^NH_3_CONH_3_^+^)^−^O-Si-O.
|           |

Such structural features of the investigated xerogels are confirmed by blue-shifted absorption peak corresponding the symmetric deformation vibrational band δ_s_(NH_2_) of urea in the FT-IR spectra (Figure 4b). The crystals of urea almost do not appear in such compositions (Figure 3a). However, an increase in urea concentration in the gel-forming system up to 10–12 wt.% promotes increased contents in urea molecules located in the space among the silicate chains.
|     H_2_O     NH_2_   H_2_O     |
O-Si-O^−^ (^+^NH_3_CONH_2_) CO (NH_2_CONH_3_^+^)^−^O-Si-O.
|     H_2_O     NH_2_   H_2_O     |

As a result, by drying the gels, the urea molecules localized in the cylindrical pores could form fibrous or needle-like crystals in the bulk of the silicate matrix (Figure 5). At the same time, in the xerogels containing more than 15 wt.% of urea, an excess of urea remains in the solution located among the gel micelles and forms large-sized regular shaped urea crystals on the surface of the resulted xerogel particles (Figure 5). The crystalline fraction of urea, estimated taking into account the intensities of the main urea reflection in the urea-containing xerogels (Figure 3a), corresponds to 6.7, 15.2, and 74.1 mol.% for the xerogels No 1, 2, and 3. 

A presence of the two types of urea crystals in the produced xerogels is also confirmed with the DSC data. A bifurcation of the endothermal peak corresponding to the melting of urea (marked as b in Figure 6b) could be explained with a presence of two kinds of urea crystals. Large crystals of urea, located on the surface of xerogel particles, melt at the temperature typical for pure crystalline urea (143 °C, Figure 6b), whereas fibrous urea crystals incorporated in the silicate matrix of urea-containing xerogels melt at lower temperatures (143, 135, and 122 °C, Figure 6 covers 4, 3, and 2, respectively) due to the influence of the silicate matrix characterized with high thermal capacity [25]. 

The resulted xerogels represent a powder consisting of particles with a size varied in the range from 0.1 to 5.0 mm. The obtained materials have the pores of 5–10 nm and 15–50 nm diameter (with a volume of 0.01 cm^3^/g and 0.17 cm^3^/g, respectively); however, the main part of pores has a closed character and appears as a result of the CO_2_ bubbling of the gel forming systems.

A presence of the pores and stable structure of xerogels based on aqueous solutions of urea in the potassium–silicate liquid glass promotes sufficiently high water resistance. As a result, the obtained xerogels containing useful chemical elements, such as K, N, as well as Na, are capable of gradual degradation of their matrix with a slow release of nutrients under the influence of aqueous solutions. This fact allows for their considering as promising controlled release fertilizers acting up to three months. 

## 4. Materials and Methods

The gelation process in the obtained mixtures was carried out in the PTFE container of viscometer (Brookfield DV-II) equipped with a spinner used as a stirrer (Figure 8). The rotation speed of the stirrer was of 30 rpm (selected taking into account the data of the preliminary experiments).

The following procedure was made to produce the xerogels based on the potassium silicate liquid glass and urea. The raw material mixtures were prepared by introduction of different quantity (100, 200, and 300 g) of powdered urea (Russian standard GOST 6691-77, purity of 99%) into 1 dm^3^ (1280 g) of the 55% aqueous solution of the potassium silicate (commercial potassium silicate liquid glass SIL-EX (PSLG) characterized with the silicate modulus equal to 3.1). The above-mentioned doses of urea were selected to obtain the final products containing, in terms of dry residue, about 5–20% of nitrogenous compound, taking into account that such concentrations are recommended for slow and controlled release fertilizers [1,2,3,4] as optimal ones. The obtained dispersion was homogenized by stirrer until the total dissolution of urea.

The gelation process taking place in the prepared mixtures was accompanied with an increase in their viscosity which was controlled by viscometer. The gelation process was considered as complete after the establishment of a constant (increased) viscosity of the obtained colloidal solutions. Further, the resulting colloidal solutions were left to mature for 2 h.

Viscosity of the potassium silicate liquid glass (PSLG) containing urea admixtures increases from 90 to 100 to 200 to1100 mPa·s depending on urea contents even without the CO_2_ bubbling. This fact confirms the participation of urea in the gelation process. However, viscosity of the colloidal solutions, produced with the admixtures of 100 and 200 g of urea into 1 dm^3^ of PSLG, was relatively low and the gels formed after maturation for 2 h settled in the aqueous solutions. That is why the solutions obtained in the systems urea-PSLG were additionally treated with CO_2_ before their maturation. Carbon dioxide (Russian standard GOST 8050–50, purity of 99.5%) was bubbled through the mixtures of urea and PSLG (colloidal solutions) using a silicon perforated tube, under a pressure of 0.2 MPa till obtaining the gels characterized with a viscosity of about 1200 mPa·s. The time of the completed gelation of the obtained colloids, taking place during the additional treatment by CO_2_ bubbling, varied from 1 to 0.25 min, decreasing with urea content. All the gels obtained after maturation of the colloids additionally treated with CO_2_ were stable and did not settle. 

To compare the structure and properties of the obtained materials, the additional gel was prepared using pure PS LG which was treated in the same reactor by CO_2_ bubbling to obtain the same viscosity of the colloid (1200 mPa·s).

Further, the obtained stable gels were dried at 50 °C for 48 h.

The Scanning Electron Microscope (SEM) Philips XL30ESEM equipped with EDS analyzer (EDAX Pegasus) was used to analyze the structural features of the obtained xerogels and to estimate their chemical composition by energy-dispersive X-ray analysis. This analysis was carried out using low electron beam voltage (5 kV), to reduce the level of errors that appear due to the volatilization of “low-weight” chemical elements (C, N).

The porosity and specific surface area of the produced xerogels were determined using the BET method with the absorption of molecular nitrogen at low temperatures. The pore size distribution was obtained by the DFT (Density Functional Theory) method using Quantachrome NOVA 200e equipment. The density of the obtained materials was measured with the Archimedes method using acetone as a liquid medium. 

The thermal stability of the obtained xerogels was investigated by DSC (Perkin Elmer), at a heating rate of 5 K/min; whereas, FT-IR spectroscopy (Nicolet Avantar 320ESP) was applied to characterize their structure. A total of 3 mg of each powdered xerogel was mixed with 300 mg of KBr and then pressed into pellets of 15 mm diameter, used to obtain IR spectra.

The phase composition of the obtained materials was carried out using a Philips PW3040 diffractometer (CuK_α_ radiation operating 40 kV and 100 mA). The reflection positions and relative intensities of the XRD patterns were compared to the catalog of the International Center for Diffraction Data (ICDD-2008).

The water resistance (chemical durability) of the xerogels obtained in the system of potassium silicate liquid glass-urea, was studied with the crushed samples (fraction of 2–4 mm). The xerogel powder (10 g) was introduced into a glass vessel with distilled water (0.5 dm^3^) at room temperature with different residence times (from 3 to 120 days) varied for each experiment. The kinetics of urea release into the aqueous solution was estimated by the photometric method in accordance with [27]. After the soaking in water for various amounts of time the obtained solutions were filtrated and their aliquots were introduced into volumetric flasks jointly with *p*-dimethylaminobenzaldehyde acetic acid, and HCl aqueous solutions and stirred. A content of urea in the analyzed aqueous solutions was estimated with a photometer KFK-3 at λ = 440 nm using the calibration data of the control experiments. 

## 5. Conclusions

The obtained results indicate that an introduction of urea in the potassium silicate liquid glass promotes fast gelation of the parent system. The structure of xerogels obtained by completed gelation of the PSLG-urea solutions with CO_2_ bubbling and following drying depends on urea content. In the gels containing about of 5 wt.% of (NH_4_)_2_CO, urea exists in the ionic form (^+^NH_3_CONH_3_^+^) and connects silicate chains supporting gelation. An increase in urea content up to 12–15 wt.% provide, in addition to the ionic form of urea, an appearance of the fibrous urea crystals incorporated into the structure of xerogel. Further increase in urea content supports increased size in urea crystals and transformation in fibrous-shaped urea into a rod-like form. A specific structure of xerogels obtained in the investigated system determines slow release of urea in water and allows for considering them as controlled released nitrogenous fertilizers containing useful nutrients (N, K). 

## Figures and Tables

**Figure 1 molecules-28-05466-f001:**
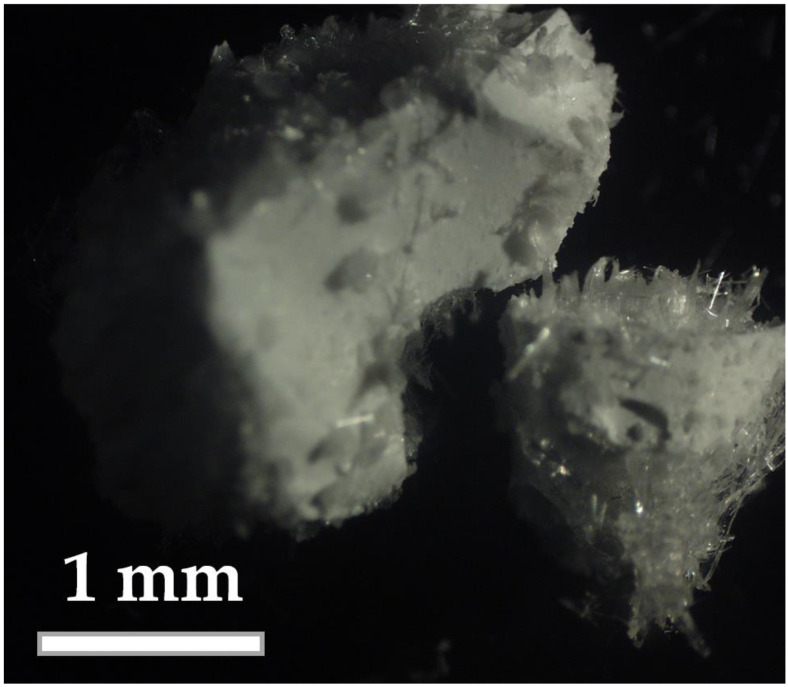
Optical image of the powder obtained by drying the gel produced in the system containing 30 g of urea in 100 mL of PSLG additionally treated with CO_2_.

**Figure 2 molecules-28-05466-f002:**
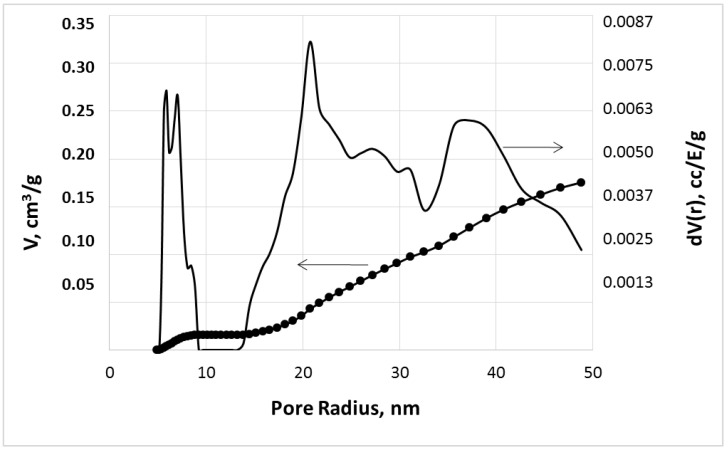
Pore size distribution and pore volume for the dried gel obtained in the system containing 30 g of urea and 100 mL of PSLG.

**Figure 3 molecules-28-05466-f003:**
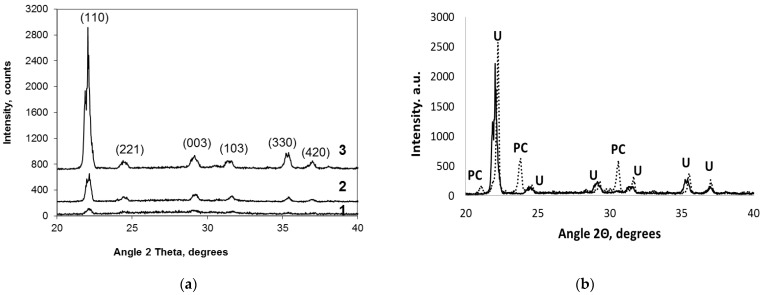
XRD patterns of the xerogels No 1, 2, and 3 (**a**), and a comparison of the XRD patterns of the xerogel NO_3_ (continuous curves) and the mechanical mixture of the urea-free xerogel, produced by CO_2_ bubbling the parent PSLG, with the same quantity (30 g) of urea additive (dotted curves) (**b**). The reflections of urea marked as U, the reflections of K_2_CO_3_ as PC (potassium carbonate).

**Figure 4 molecules-28-05466-f004:**
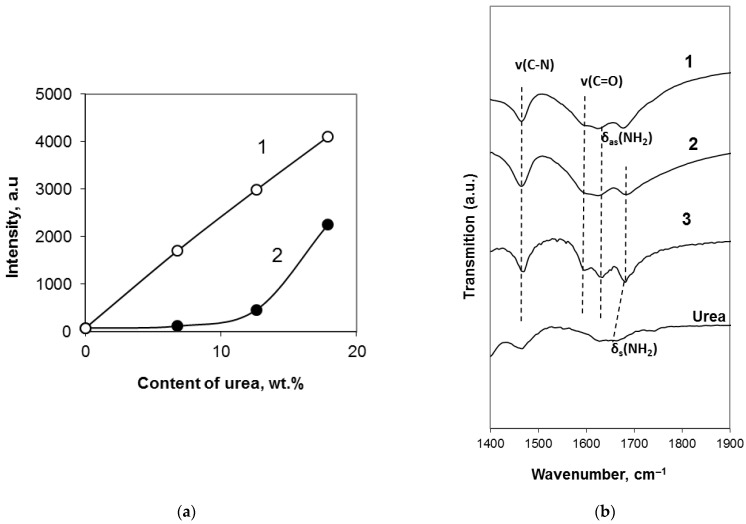
(**a**) Dependence of the integrated intensity of the crystalline urea main reflection in the XRD patterns of the xerogels produced with different admixtures of urea. 1—mechanical mixtures of urea-free xerogel and powdered urea (System 1); 2—xerogels based on urea containing gels (System 2). (**b**) FT-IR spectra in the range of urea related absorption bands.

**Figure 5 molecules-28-05466-f005:**
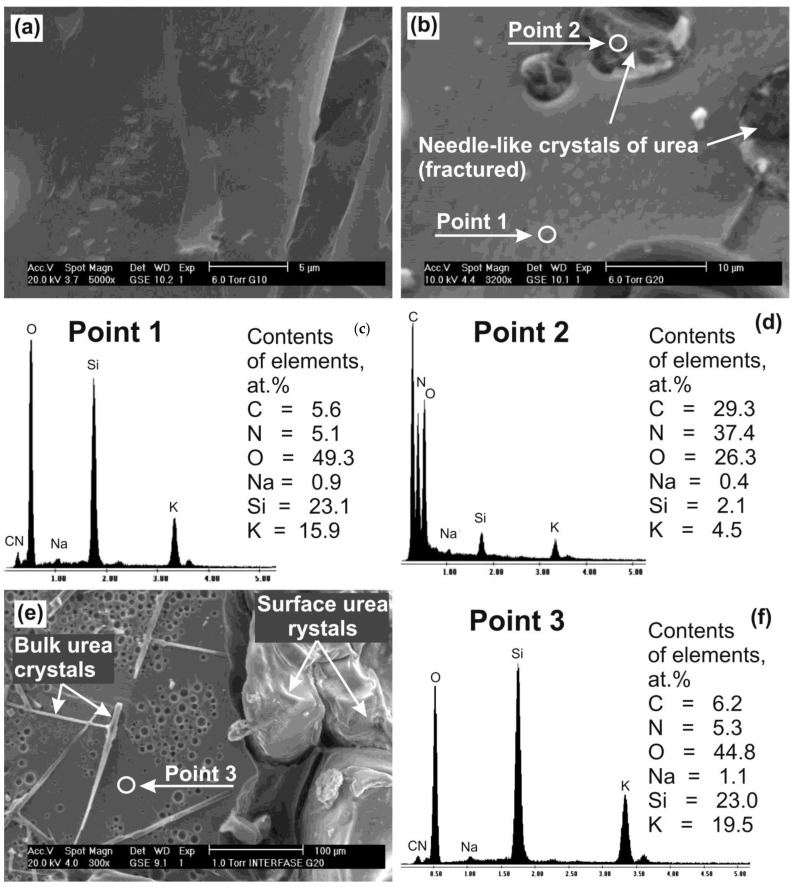
SEM images of the xerogels (fractured specimens) containing 6.8 (**a**), 12.6 (**b**), and 17.9 (**e**) wt.% of urea and their local XRF spectra obtained in the points marked in the images (**b**,**e**) as well as the corresponding chemical composition estimated by low energy EDS analysis (**c**,**d**,**f**).

**Figure 6 molecules-28-05466-f006:**
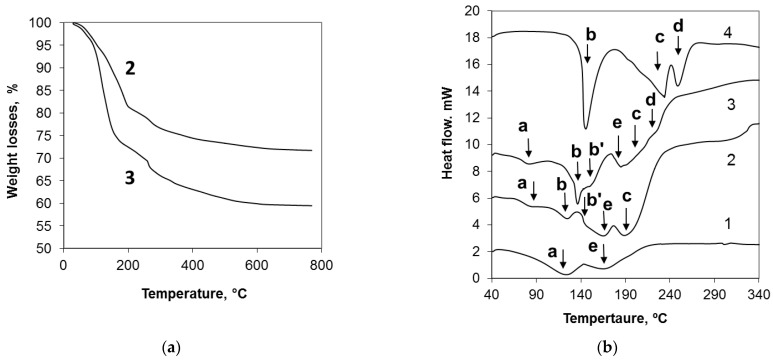
TGA (**a**) and DSC (**b**) covers of the xerogels based on potassium silicate liquid glass treated with CO_2_ (cover 1), xerogels No 2 and 3 (Table 1) based on the system of PSLG-urea (curves 2 and 3) and pure urea (curve 4). The arrows and letters indicate various thermal effects described in the text.

**Figure 7 molecules-28-05466-f007:**
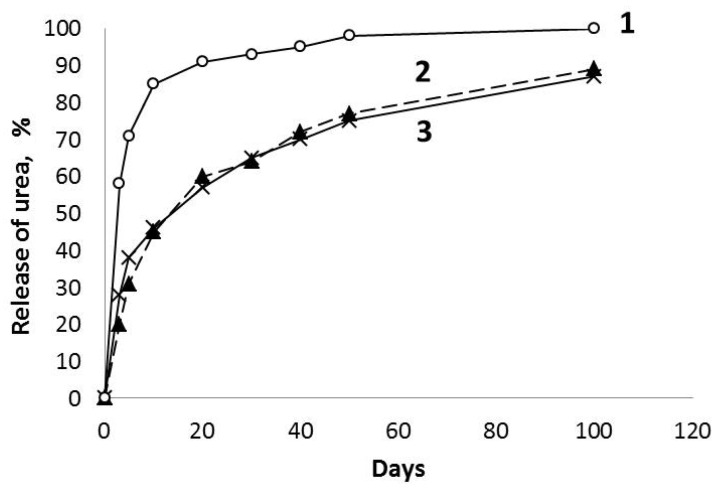
Release of urea in water (kinetic covers) for the mechanical mixture of the commercial crystalline urea (17.9 wt.%) and the xerogel (fraction of 2–4 mm) based on urea-free PSLG treated by CO_2_ (1) in comparison with the xerogels of the same fraction containing 12.6 (2) and 17.9 (3) wt.% of urea.

**Figure 8 molecules-28-05466-f008:**
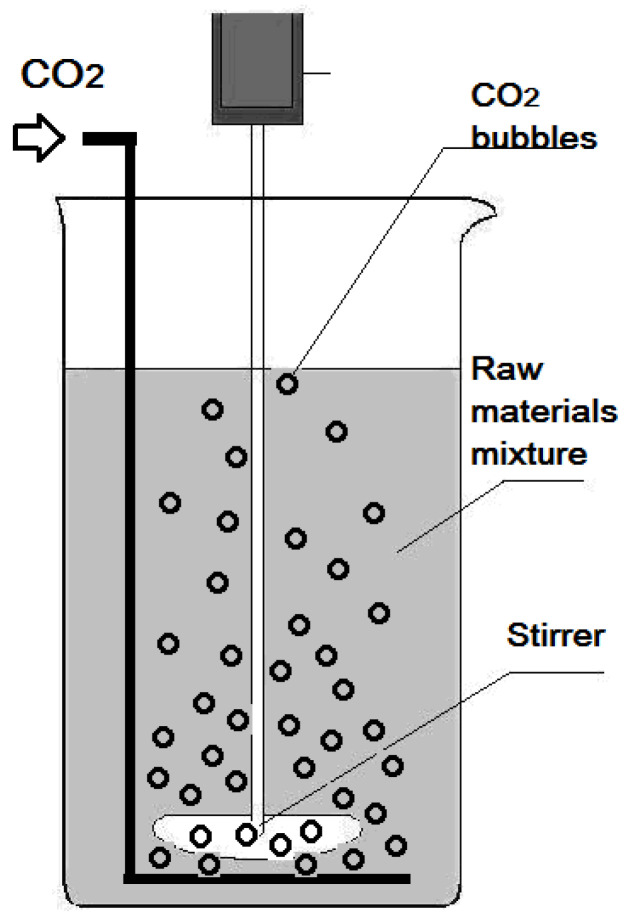
The scheme of a reactor used to produce the gels in the investigated system.

**Table 1 molecules-28-05466-t001:** Chemical composition and porosity of the xerogels based on the PSLG with different admixtures of urea treated by CO_2_ bubbling until completed gelation (estimated from the raw material compositions).

No of the Gel	Admixture of Urea to 100 mL of PSLG, g	Contents of the Components, wt.%			Density, g/cm^3^	Specific Surface Area, m^2^/g	Porosity, %
		K_2_O·3.1SiO_2_	Urea	H_2_O			
1	10	83.8	5.8	10.4	1.19 ± 0.01	158	20.6
2	20	75.2	12.6	12.2	1.17 ± 0.02	145	18.7
3	30	63.8	17.9	18.3	1.14 ± 0.02	139	17.9

## Data Availability

Not applicable.

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
