# Peer review of "Structure and Properties of the Xerogels Based on Potassium Silicate Liquid Glass and Urea"

_molecules, 2023, doi:10.3390/molecules28145466_

Round 1

Reviewer 1 Report

The paper entitled "Structure and properties of the aerogels based on potassium silicate liquid glass and urea" is adequately organized and uses methodology that corresponds to this type of research.

However, the authors must introduce some changes, which are mentioned below:

1. The description of aerogel synthesis should be greatly improved, with reactions, specific conditions (and why they were chosen) and scheme of the reactor used.

2. The authors will present a textural and chemical characterization of the aerogels. This with the aim of correlating it with the synthesis route.

3. Propose mechanisms and justify them through spectroscopic techniques.

4. The authors should be clearer when explaining the mixtures in Table 1. Is there any experimental design to establish these mixtures?

5. In Figure 2, the DRX must be accompanied by the respective planes (Miller indices).

6. Figure 3 on crystallinity should be better explained.

7. The resolution of Figure 4 needs to be improved.

8. The explanation of the results of the DSC is not clear. Please expand and improve.

9. Physical tests such as hardness, electrical conduction etc. are suggested to be carried out.

10. The bibliographical references should be expanded.

Dear

Editor

This research is of interest and has new aspects to the state of the art of aerogels. However the authors need to do more work on the paper and I have suggested a number of changes. If they introduce it, I will agree with their publication.

Sincerely yours

Author Response

Dear colleague, thank you very much for your recommendations on our manuscript. We have tried to meet all your wishes in the corrected version of the paper (enclosed). Some our comments and answers are reported below.

  1. The description of aerogel synthesis should be greatly improved, with reactions, specific conditions (and why they were chosen) and scheme of the reactor used.

The reason to use urea as a gelating agent is described in the Introduction part. The experimental conditions to produce the gels (xerogels) were selected taking into account the results of the preliminary experiments described in the part 2.1. The description of the experimental conditions and rector used to obtain the gels have been extended.

  1. The authors will present a textural and chemical characterization of the aerogels. This with the aim of correlating it with the synthesis route.

The description of textural and chemical characteristics of the obtained xerogels is extended too.  A term “xerogel” was used in the corrected version of the manuscript instead of a term “aerogel”, taking into account the recommendation of another reviewer.

  1. Propose mechanisms and justify them through spectroscopic techniques.

The mechanism of the processes accompanied gelation and drying the gels is considered with more details.

  1. The authors should be clearer when explaining the mixtures in Table 1. Is there any experimental design to establish these mixtures?

The compositions reported in Table 1 was not designed. They correspond to the material balance taking into account the data on drying weight loss. The reasons for choosing just such compositions of raw material mixtures have been given in Section 4.

  1. In Figure 2, the DRX must be accompanied by the respective planes (Miller indices).

Done

  1. Figure 3 on crystallinity should be better explained.

Done

  1. The resolution of Figure 4 needs to be improved.

New FT-IR spectra were obtained and discussed

  1. The explanation of the results of the DSC is not clear. Please expand and improve.

Done

  1. Physical tests such as hardness, electrical conduction etc. are suggested to be carried out.

We did not make the mechanical and electrical testing. Taking into account that the obtained xerogels are proposed to be used as the controlled release fertilizers, we think that such measurements would not present any information useful for such application.

  1. The bibliographical references should be expanded.

Done

Reviewer 2 Report

This work is not recommended for publication in Molecules.

1.     The samples obtained in this work is xerogels, not aerogels.

2.     The characterization of this work is very inadequate, and tests such as BET, which are important for aerogel, have not been carried out.

3.     The figures in the manuscript are unattractive.

Author Response

Dear colleague, thank you very much for your recommendations on our manuscript. We have tried to meet your wishes in the corrected version of the paper (enclosed). Some our comments and answers are reported below.

1. The samples obtained in this work is xerogels, not aerogels.         

We agree. The corresponding collections were made in all the manuscript.

2. The characterization of this work is very inadequate, and tests such as BET, which are important for aerogel, have not been carried out

The data obtained using the BET technique were introduced and discussed  in the manuscript

3. The figures in the manuscript are unattractive.       

We have tried to improve a quality of Figures

Reviewer 3 Report

1.       Some paragraphs are too short. Please give more descriptions and explanations.

2.       As for the FT-IR spectrum, please use to the raw data and re-draw the figure to be clear to the readers.

3.       It is recommended to add some relative literatures such as Ceramics International, 2016, 42(1): 874-882; Ceramics International, 2018, 44(1): 821-829.

4.       English writing and grammar should be carefully checked throughout the manuscript.

5.       A feature of aerogels is the three-dimensional porous structures, please provide the N2-adsroption and desorption test results for the analysis of BET specific surface area and pore size distributions.

6.       The units of urea added in the mixture should be provides in Table 1.

7.       The captions in Figure. 4 do not keep consistent with the curves in the figure.

Moderate editing of English language required

Author Response

Dear colleague, thank you very much for your recommendations on our manuscript. We have tried to meet all your wishes in the corrected version of the paper (enclosed). Some our comments and answers are reported below:

  1. Some paragraphs are too short. Please give more descriptions and explanations.

The text of the manuscript was extended with more descriptions and explanations.

  1. As For the FT-IR spectrum, please use to the raw data and re-draw the figure to be clear to the readers.

Done

  1. It is recommended to add some relative literatures such as Ceramics International, 2016, 42(1): 874-882; Ceramics International, 2018, 44(1): 821-829.

Done

  1. English writing and grammar should be carefully checked throughout the manuscript.

Done

  1. A feature of aerogels is the three-dimensional porous structures, please provide the N2-adsroption and desorption test results for the analysis of BET specific surface area and pore size distributions.

Taking into account the recommendation of another reviewer, a term “xerogel” was used in the corrected version of the manuscript instead of a term “aerogel”. 

  1. The units of urea added in the mixture should be provides in Table 1.

Done

  1. The captions in Figure. 4 do not keep consistent with the curves in the figure.

The captions were corrected.

Round 2

Reviewer 2 Report

All issues of concern have been responded to and revised.